# Spatial patterns of tuberculosis and HIV co-infection in Ethiopia

**Kefyalew Addis Alene**[1,2]*, **Kerri Viney**[3], **Hannah C. Moore**[2], **Maereg Wagaw**[4], **Archie C. A. Clements**[1,2]

**1** Faculty of Health Sciences, Curtin University, Perth, Western Australia, Australia, **2** Wesfarmers Centre of Vaccines and Infectious Diseases, Telethon Kids Institute, Perth, Western Australia, Australia, **3** Research School of Population Health, College of Health and Medicine, The Australian National University, Canberra, Australian Capital Territory, Australia, **4** Ethiopia Federal Ministry of Health, Addis Ababa, Ethiopia

* kefyalew.alene@anu.edu.au

**Data Availability Statement:** All relevant data are within the paper and its Supporting Information files.

**Funding:** The author(s) received no specific funding for this work.

## Abstract

### Background

Tuberculosis (TB) and human immunodeficiency virus (HIV) are the leading causes of infectious-disease-related deaths in Ethiopia, but little is known about their spatial distribution across the country. The aim of this study was to investigate the spatial patterns of TB and HIV co-infection in Ethiopia at the district level.

### Methods

We conducted an ecological study using TB and HIV data reported from all regions of Ethiopia through the national Health Management Information System (HMIS), between June 2015 and June 2017. Spatial clustering was assessed using Moran's I statistic and Getis-Ord statistic. Spatial binomial regression models were constructed separately for the prevalence of TB among people living with HIV and for the prevalence of HIV among TB patients, with and without spatial components using a Bayesian approach.

### Results

A total of 1,830,880 HIV and 192,359 TB patients were included in the analysis. The prevalence of HIV among TB patients was 7.34%; hotspots were observed in districts located in Amhara, Afar, and Gambela regions, and cold spots were observed in Oromiya and Southern Nations, Nationalities, and People (SNNP) regions. The prevalence of TB among people living with HIV varied from 0.7% in Oromia region to 14.5% in Afar region. Hotspots of TB prevalence among people living with HIV were observed in districts located in Gambela, Afar, Somali, and Oromiya regions; whereas the cold spots were observed in districts located in Amhara and Tigray regions. The ecological-level factors associated with the prevalence of TB among people living with HIV were low wealth index (OR: 1.49; 95% CrI: 1.05, 2.05), low adult literacy rate (OR: 0.67; 95% CrI: 0.46, 0.94), and distance to an international border (OR: 0.61; 95% CrI: 0.40, 0.91). The factors associated with the prevalence of HIV among TB patients were poor health care access (OR: 0.76; 95% CrI: 0.59, 0.95), low

**Competing interests:** The authors have declared that no competing interests exist.

wealth index (OR: 1.31; 95% CrI: 1.01, 1.67), and low adult literacy rate (OR: 1.37; 95% CrI: 1.03, 1.78).

## Conclusion

Our study provides evidence for geographic clustering of TB/HIV co-infection in Ethiopia. Health care access, proximity to international borders, and demographic factors such as low wealth index and adult literacy were significantly associated with the prevalence of TB/HIV co-infection.

## Introduction

Tuberculosis (TB) and human immunodeficiency virus (HIV) are the first and the second leading cause of death attributable to infectious diseases worldwide, respectively. The synergy between TB and HIV infection is strong. While TB is the leading cause of death among people living with HIV [1], HIV infection is the most important risk factor for developing active TB [2, 3]. Tuberculosis and HIV co-infection has been a major public health challenge to the global efforts in achieving elimination goals for both diseases. There were an estimated 36 million people living with HIV globally in 2017, with approximately one third of these people being co-infected with TB [4]. The highest burden of TB/HIV co-infection is reported from sub-Saharan Africa [4].

Ethiopia is one of the 30 high TB- and HIV-burden countries in the world [5–7], with incidence of 0.17 cases per 1000 population for HIV and 1.64 per 1000 for TB [8–10]. Studies conducted in Ethiopia have shown that TB/HIV co-infected individuals have a greater risk of psychosocial problems, low quality of life, and poorer physical health than HIV infected individuals without active TB [11–13]. The prevalence of TB/HIV co-infection has been found to vary widely in Ethiopia [14, 15] and other countries [16–18], partially because of differences in health care access and other socio-demographic factors such as wealth index and literacy rate.

Although several small scale studies have reported the epidemiology of TB/HIV co-infection in many parts of Ethiopia [19, 20], to our knowledge, there are no reports on the spatial distribution of TB-HIV co-infection nationally. Identifying areas where the burden of TB/HIV co-infection is concentrated might help identify populations at higher risk of co-infection, and knowledge of high and low burden areas is required for successful surveillance programs and optimal resource allocation [21], including identifying areas where existing TB and HIV collaborative programs and integrated services need to be strengthened. Therefore, this study aimed to assess the spatial distribution of TB/HIV co-infection in Ethiopia at the district level and investigate ecological-level factors associated with the prevalence of TB among people living with HIV and the prevalence of HIV among TB patients.

## Methods

### Study design and setting

An ecological study was conducted in Ethiopia. Our study estimated the prevalence of TB/HIV co-infection to the district level, using the national TB/HIV case notification data, reported from June 2015 to June 2017. The district is the third administrative level in Ethiopia. Districts comprise a well-defined population within a clearly delineated administrative and geographical area. Districts contain networks of primary health care units such as health centers, health posts and a district hospital. Health related data are compiled at the district level

and are reported to the Zone Health Department, Region Health Bureau and then to the Federal Ministry of Health. Both TB and HIV are notifiable diseases in Ethiopia. These diseases are reported by the district Health Office to the Federal Ministry of Health through the Health Management Information System (HMIS) on a quarterly basis [22, 23]. The data include the number of: TB patients enrolled into the national TB program, TB cases who were tested for HIV, TB patients with HIV positive test results; HIV positive clients who received HIV care during the reporting period; HIV positive clients who were screened for TB during their visit; and HIV positive clients with active TB. TB and HIV diagnosis and treatment are integrated and are provided free of charge in Ethiopia across the country.

## Data variables

The dependent variable in this study was numbers of TB and HIV co-infection in two study populations: people living with HIV and TB patients. This was determined as the prevalence of TB among people living with HIV and the prevalence of HIV among TB patients. Individual data on infection status with TB and HIV for the entire population or a representative sample were not available, precluding the use of a single (multinomial) model incorporating all possible combinations of TB and HIV mono- and co-infections.

All TB/HIV co-infected individuals reported to the national HMIS during the study period were included in the study. Study participants whose current address of residence was not registered, or unknown were excluded from the analysis because their data was unable to be georeferenced to the district level. The independent variables were socio-economic factors (such as low wealth index, average number of persons per room, the unemployment rate, and the adult literacy rate), behavioural factors (such as chat chewing and alcohol drinking), poor health care access (i.e. difficulty of getting advice or treatment due to lack of money, or distance to a health facility), distance to an international border (i.e. straight-line distance to the nearest international border), and climatic factors (such as mean temperature, precipitation, and aridity). A table showing the summary of independent variables, their sources of data and units of measurement is presented as supplementary information (S1 Table). These independent variables were extracted from the Ethiopian Demographic and Health Survey (EDHS 2011 and EDHS 2016), the Ethiopia Atlas of Population Density, and EDHS Spatial Data. The variables were selected a priori due to their plausible association with the prevalence of HIV/TB co-infection.

A Polygon shapefile for Ethiopia's administrative boundaries at the district level was obtained from the Open Africa website (https://africaopendata.org/group/ethiopia). The TB and HIV datasets and the study covariates were merged with the district level polygons using ArcGIS (ESRI, Redlands, CA) geographical information system (GIS) software [24].

## Data analysis

Descriptive statistics and geospatial analyses were conducted. The prevalence of TB among people living with HIV was calculated as the number of TB patients with an HIV positive test result in a district divided by the total number of clients enrolled in HIV care who were screened for TB during their visit in the same district. Similarly, the prevalence of HIV among TB patients was calculated as the number of TB patients with an HIV positive test result in a district divided by the total number of TB patients who were tested for HIV.

## Spatial analysis

Spatial autocorrelation (based on the feature locations and attribute values) in prevalence of co-infection was measured using the Global Moran's I statistic in order to assess the overall

pattern and trend of the data. Local spatial clustering was explored using the Getis-Ord statistic. An initial candidate set of demographics, behavioural, climatic and environmental covariates were considered for inclusion in the spatial models to investigate the ecological-level factors associated with the clustering of TB and HIV co-infection prevalence. Variable selection was made using fixed-effects univariable regression models and all variables with P<0.2 were selected for inclusion in final multivariable models. Variables were also checked for the presence of multi-collinearity, and those variables with a variance inflation factor greater than six were excluded from the final spatial model. The residuals of the final models were examined for the presence of spatial autocorrelation.

Spatial binomial regression models were constructed separately for the prevalence of TB among people living with HIV and the prevalence of HIV among TB patients in WinBUGS version 1.4 (MRC Biostatistics Unit, Cambridge, and Imperial College London, UK). Models were developed with and without spatial components using a Bayesian approach.

We constructed the model with a spatially structured and unstructured random effects. We assumed that the observed prevalence of TB among people living with HIV (r) at zone (i) had a binomial distribution with a total number of TB patients who were tested for HIV at the zone (ni) and the modelled prevalence of TB among people living with HIV at the zone (pi): ri ~ Binomial (ni, pi). For this model, pi was specified as follows: $logit\,(p_i) = \alpha + \Sigma_N \beta_m{}^* X_{n,i} + U_i + V_i$ where $\alpha$ is the intercept; $\Sigma_N \beta_m{}^* X_{n,i}$ is the matrix of $m$ independent zone-specific variables X (i.e. proportion of the population with a low wealth index, alcohol drinking, poor health care access, adult literacy rate, and distance to an international border) measured at each zone i, multiplied by their coefficients ($\beta$). $U_i$ are unstructured random effects; and $V_i$ are the spatially structured random effects, modelled using a conditional autoregressive (CAR) approach. The CAR component was defined using an adjacency matrix to determine the spatial relationships between zones. The adjacency matrix for each zone was generated using ArcGIS based on the queen definition, whereby two areas are considered neighbours if they share a common boundary or vertex. A weight of 1 was given if two zones were neighbours and 0 if otherwise. The posterior parameters were estimated using a Bayesian Markov Chain Monte Carlo (MCMC) simulation. Non-informative priors were used for $\alpha$ (uniform prior with bounds —∞ and +∞) and the coefficients (normal prior with mean = 0 and precision 1× 10−6). The priors for the precision of the unstructured and spatially structured random effects were given non-informative gamma distributions with shape and scale parameters set at 0.001. The deviance information criterion (DIC) statistic was calculated to select the best-fitting model (models with a lower DIC statistic are considered to show a better compromise between model fit and parsimony). Convergence of the parameter values was observed using visual examination of history and density plots of the model chains. Convergence of the models was successfully achieved after 400,000 iterations. The models were run for a further 100,000 iterations and the posterior values were stored after 500,000 iterations. Using these procedures and steps, similar but separate models were constructed for the prevalence of HIV among TB patients. Models were also developed for each outcome excluding either the unstructured or structured random effects.

## Ethics clearance

This study was approved by the Australian National University Human Research Ethics Committee (protocol number 2016/218) and a letter of endorsement was obtained from the Ethiopian Ministry of Health. As this study used district-level aggregated data, informed consent was not obtained from the study participants.

**Table 1. The number of TB, HIV and TB/HIV patients reported in Ethiopia, 2015–2017.**

| Variable | Male | Female | Total |
|---|---|---|---|
| Total number of HIV positive clients who received HIV care during the reporting period | 701,963 | 1,128,917 | 1,830,880 |
| Number of clients enrolled in HIV care who were screened for TB during their visit | 647,807 | 1,037,496 | 1,685,303 |
| Total number of HIV positive clients with active TB | 7,531 | 6,621 | 14,152 |
| Number of TB cases enrolled to DOTS and who were tested for HIV | 105,387 | 86,972 | 192,359 |
| Number of TB patients with HIV positive test result | 7,472 | 6,716 | 14,188 |
| Number of TB patients with HIV Negative test result | 96,917 | 79,589 | 176,506 |
| Number of TB patients with unknown or undocumented HIV test result | 998 | 667 | 1,665 |

HIV: Human Immunodeficiency Virus; TB: Tuberculosis; DOTS: Directly Observed Therapy, Short Course.

## Results

A total of 1,830,880 HIV and 192,359 TB patients reported to the national HMIS from June 2015 to June 2017 were included in this study. These data covered all zones and regions of Ethiopia, except for 23 districts that had no TB or HIV data available. From the total 1,830,880 HIV positive clients who received HIV care in Ethiopia during the study period, 1,685,303 (92%) were screened for TB. A higher proportion of clients enrolled in HIV care who were screened for TB during the study period were female (61.6%; n = 1,128,917) than male (38.4%; n = 701,963), whereas a higher proportion of TB cases enrolled to Directly Observed Therapy Short Course (DOTS) and who were tested for HIV were male (54.8%; n = 105,387) than female (45.2%; n = 86,972). Table 1 presents the total number of TB and HIV patients in Ethiopia, stratified by sex. The overall prevalence of TB among people living with HIV was 0.84%; and varied greatly by sex, with a higher TB prevalence found among males (1.2%) than females (0.64%). The prevalence of HIV among TB patients was 7.34% and was slightly lower (7.09%) for males than females (7.72%). The highest prevalence of TB among people living with HIV was reported in Somalia (14.5%), Gambela (9.6%), and Afar (8.5%) regions (Table 2; Fig 1). The highest prevalence of HIV among TB patients was reported in Addis Ababa (22.5%), and the lowest prevalence was reported in Southern Nations, Nationalities, and People (SNNP) (3.8%) and Oromia (4.8%) regions (Table 2; Fig 2). The prevalence of TB among people living with HIV and the prevalence of HIV among TB patients at country, region, and zone levels are provided in the supplementary tables (S2 Table).

**Table 2. The prevalence of TB among HIV patients, and the prevalence of HIV among TB patients in all regions of Ethiopia.**

| | TB prevalence among HIV patients (%) | HIV prevalence among TB patients (%) |
|---|---|---|
| Addis Ababa | 1.0 | 22.5 |
| Afar | 8.5 | 8.1 |
| Amhara | 0.6 | 8.2 |
| Beneshangul Gumuz | 7.7 | 6.3 |
| Dire Dawa | 1.0 | 12.6 |
| Gambela | 9.6 | 20.8 |
| Harari | 0.9 | 8.9 |
| Oromiya | 0.7 | 4.8 |
| SNNP | 1.6 | 3.8 |
| Somali | 14.5 | 5.5 |
| Tigray | 0.9 | 8.4 |
| Ethiopia | 0.8 | 7.4 |

## Spatial clustering of TB/HIV co-infection

There was evidence of spatial clustering of TB/HIV co-infection, with a global Moran's I = 0.13 (p value < 0.001) for the prevalence of TB among people living with HIV and 0.17 (p value < 0.001) for the prevalence of HIV among TB patients. The Getis-Ord statistics showed that the prevalence of HIV among TB patients in Ethiopia was clustered in districts located in Amhara region and the prevalence of HIV among TB patients in the same region is relatively wide-spread. Hotspots of TB prevalence among people living with HIV were observed in districts located in Gambela, Afar, Somali, and Oromiya regions; whereas the cold spots were observed

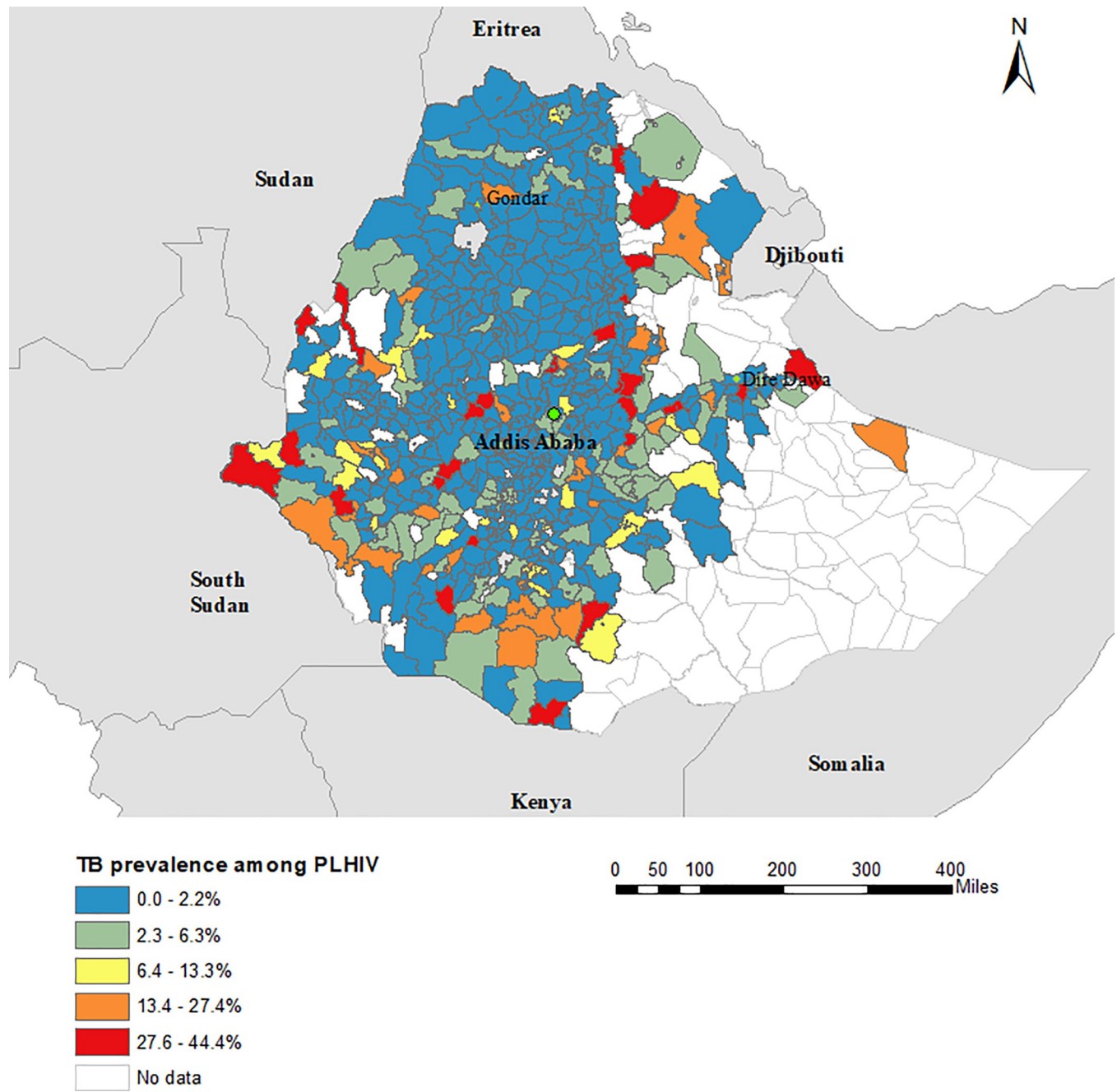

**Fig 1. Geographical distribution of the prevalence of tuberculosis among people living with HIV (PLHIV) in Ethiopia, June 2016 and June 2017.**

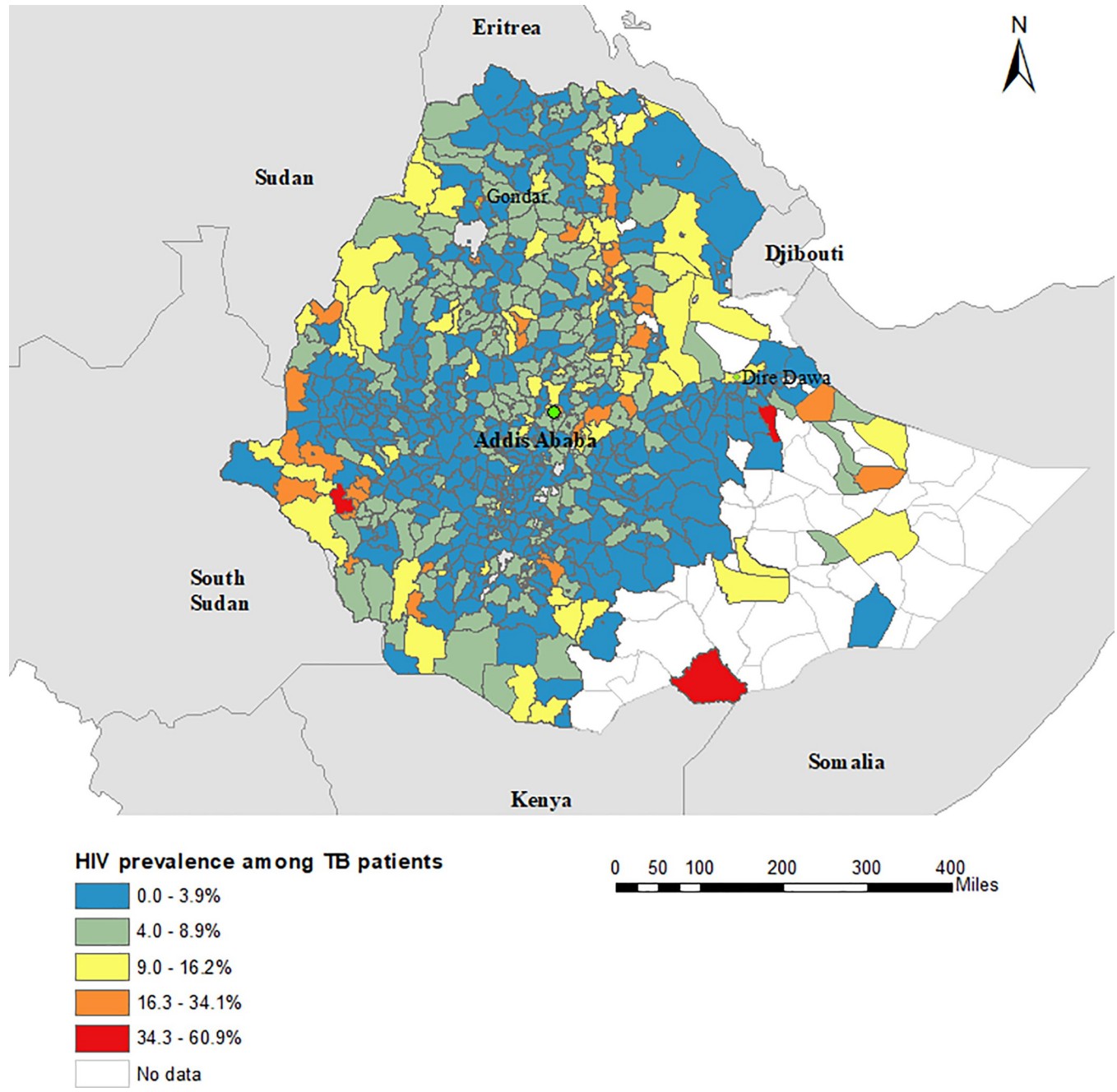

**Fig 2. Geographical distribution of the prevalence of HIV among tuberculosis patients in Ethiopia, June 2016 and June 2017.**

in districts located in Amhara and Tigray regions (Fig 3). For the prevalence of HIV among TB patients, hotspots were observed in districts located in Amhara, Afar, and Gambela regions, and cold spots were observed in Oromiya and SNNP regions (Fig 4).

## Factors associated with spatial clustering of TB/HIV co-infection

Table 3 shows the univariate models fitted for both the prevalence of TB among people living with HIV and the prevalence of HIV among TB patients. For both outcomes, the best-fitting

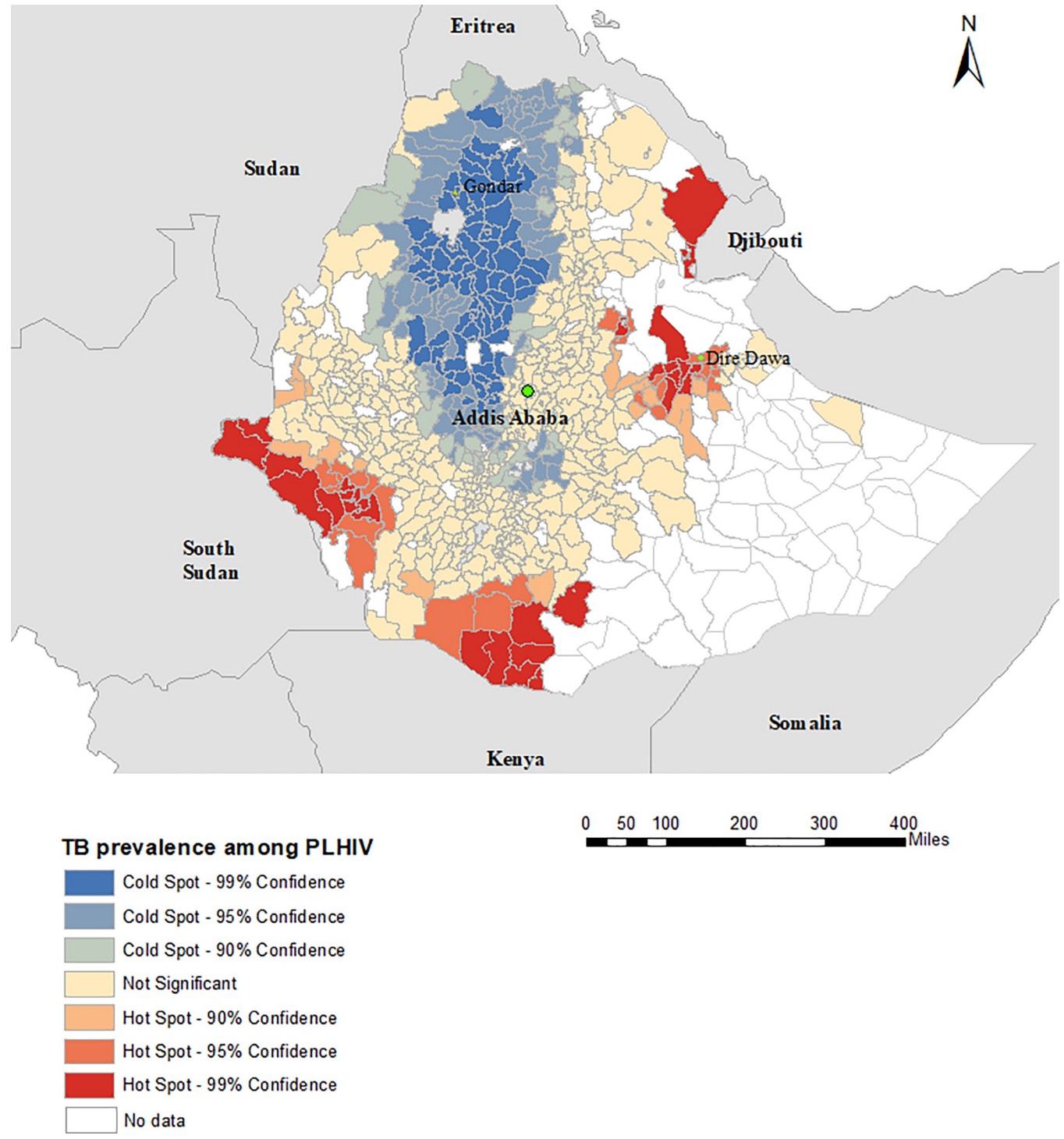

**Fig 3. Spatial clustering of TB prevalence among people living with HIV (PLHIV) at district-level in Ethiopia, 2015 to 2017, based on the Getis-Ord Gi\* statistic.**

model was the one with spatially structured random effects, but no spatially unstructured random effects. In the best-fitting model for the prevalence of TB among people living with HIV, low wealth index (odds ratio (OR): 1.49; 95% CrI: 1.05, 2.05), low adult literacy rate (OR: 0.67;

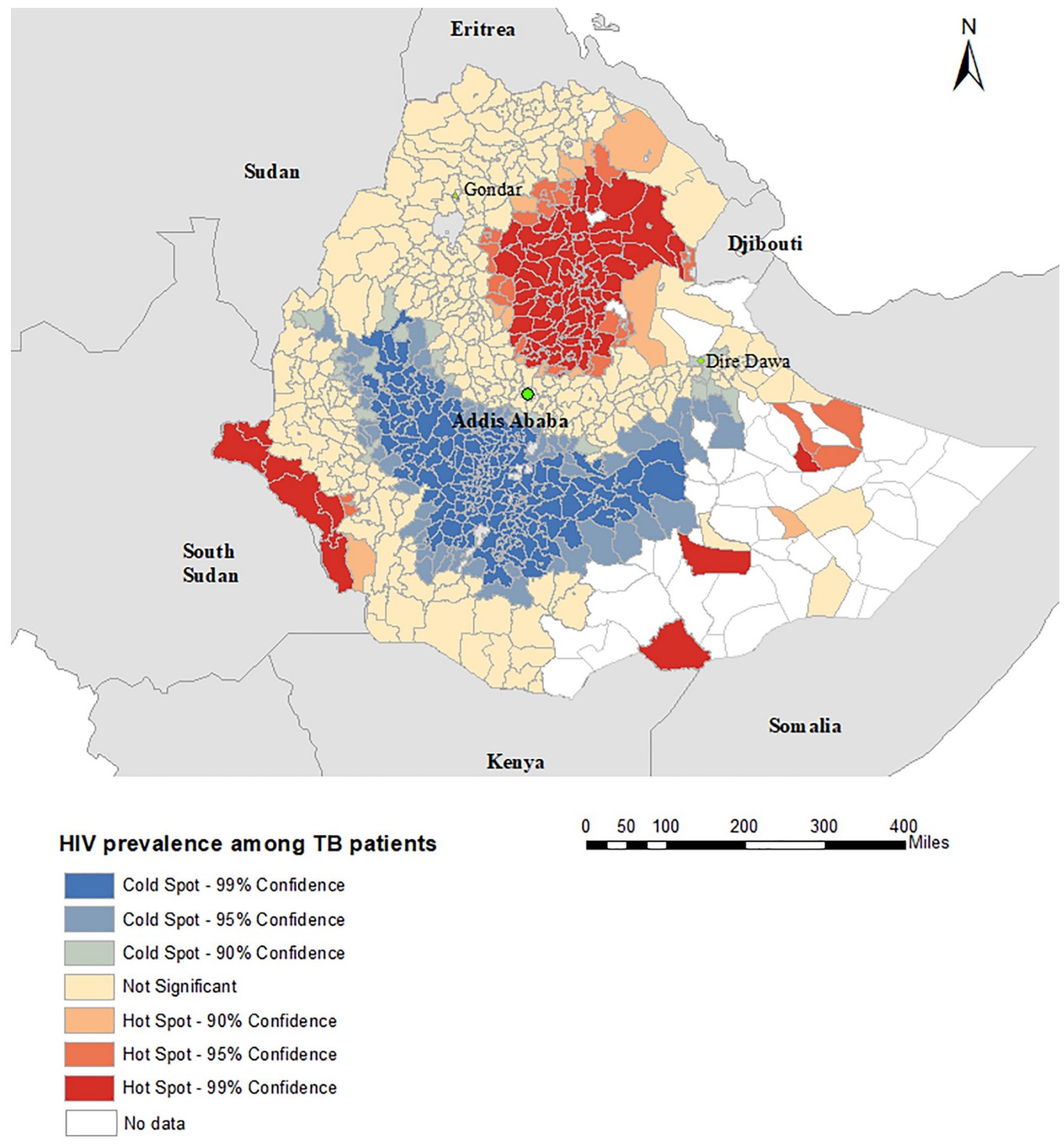

**Fig 4. Spatial clustering of HIV prevalence among TB patients at district-level in Ethiopia, 2015 to 2017, based on the Getis-Ord Gi* statistic.**

95% CrI: 0.46, 0.94), and proximity (distance) to international borders (OR: 0.61; 95% CrI: 0.40, 0.91) were significantly associated with TB (Table 4). When the model was fitted with the prevalence of HIV among TB patients, poor health care access (OR: 0.76; 95% CrI: 0.59, 0.95),

**Table 3. Univariate model for prevalence of TB among HIV patients, and prevalence of HIV among TB patients in Ethiopia, 2015–2017.**

| | TB prevalence among HIV patients | | HIV prevalence among TB patients | |
|---|---|---|---|---|
| Independent variables | Coefficient (95% CI) | P-value | Coefficient (95% CI) | P-value |
| Low wealth index | 0.12 (0.06, 0.18) | 0.00 | 0.01 (-0.06, 0.09) | 0.19 |
| Average number of persons per room | 2.32 (0.70, 3.95) | 0.006 | 0.73 (-0.81, 2.29) | 0.34 |
| Unemployed population | -0.123 (-0.46, 0.21) | 0.463 | 0.57 (0.22, 0.93) | 0.002 |
| Adult literacy rate | -0.14 (-0.24, -0.04) | 0.004 | 0.04 (-0.05, 0.15) | 0.15 |
| Chat chewing | -0.007 (-0.09, 0.08) | 0.868 | -0.02 (-0.12, 0.08) | 0.69 |
| Alcohol drinking | -0.06 (-0.11, -0.007) | 0.027 | -0.02 (-0.08, 0.04) | 0.18 |
| Health care access problem | 0.044 (-0.043, 0.13) | 0.19 | -0.09 (-0.19, 0.003) | 0.058 |
| Poor knowledge towards TB | 0.13 (0.007, 0.26) | 0.038 | 0.05 (-0.09, 0.19) | 0.49 |
| Poor attitude towards TB | 0.104 (0.005, 0.203) | 0.039 | -0.05 (-0.17, 0.06) | 0.34 |
| Enhanced vegetation index | 0.007 (-0.01, 0.01) | 0.932 | -0.008 (-0.03, 0.009) | 0.34 |
| Rainfall | -0.0005 (-0.002, 0.001) | 0.54 | 0.01 (-0.03, 0.01) | 0.24 |
| Aridity | -0.005 (-0.01, 0.0009) | 0.098 | -0.005 (-0.01, 0.001) | 0.10 |
| Mean temperature | 1.095 (0.63, 1.55) | 0.001 | 0.53 (0.008, 1.05) | 0.04 |
| Proximity to international border | -0.02 (-0.03, -0.09) | 0.001 | -0.012 (-0.028, 0.06) | 0.12 |

low wealth index (OR: 1.31; 95% CrI: 1.01, 1.67), and low adult literacy rate (OR: 1.37; 95% CrI: 1.03, 1.78), were significantly associated with HIV prevalence (Table 4). When the residuals of the final model were examined for spatial autocorrelation, after incorporating the socioeconomic and health care access variables into the model, spatial autocorrelation was apparent (Global Moran's index = 0.09; p-value = 0.04), indicating that the covariates were not able to

**Table 4. Bayesian logistic regression model for the association of ecological level covariates and spatially structured and unstructured random effect with the prevalence of TB among HIV patients and prevalence of HIV among TB patients in Ethiopia, 2015 to 2017.**

| | TB prevalence among HIV patients | | | HIV prevalence among TB patients | | |
|---|---|---|---|---|---|---|
| | Spatial unstructured | Spatially structured | Both spatially structured and unstructured | Spatial unstructured | Spatially structured | Both spatially structured and unstructured |
| Independent variables | OR (95% CI) | OR (95% CI) | OR (95% CI) | OR (95% CI) | OR (95% CI) | OR (95% CI) |
| Low wealth index | 1.53 (1.08, 2.10) | 1.49 (1.05, 2.05) | 1.53 (1.08, 2.10) | 1.24 (0.94, 1.59) | 1.31 (1.01, 1.67) | 1.31 (1.01, 1.67) |
| Adult literacy rate | 0.866 (0.62, 1.18) | 0.67 (0.46, 0.94) | 0.80 (0.55, 1.11) | 1.27 (0.98, 1.62) | 1.37 (1.03, 1.78) | 1.36 (1.03, 1.78) |
| Alcohol drinking | 0.69 (0.55, 1.85) | 0.74 (0.55, 1.98) | 0.71 (0.53, 0.92) | 0.99 (0.83, 1.19) | 1.10 (0.87, 1.37) | 1.08 (0.84, 1.36) |
| Poor health care access | 0.86 (0.67, 1.10) | 0.80 (0.59, 1.07) | 0.78 (0.58, 1.02) | 0.78 (0.64, 0.95) | 0.76 (0.59, 0.95) | 0.74 (0.58, 0.93) |
| Proximity to international border | 0.71 (0.55, 0.90) | 0.61 (0.40, 0.91) | 0.74 (0.53, 1.00) | 0.90 (0.74, 1.09) | 0.77 (0.55 (1.03) | 0.82 (0.61, 1.08) |
| | | | | | | |
| Intercept | -4.02 (-4.24, -3.80) | -4.06 (-4.16, -3.96) | -4.03 (-4.22, -4.03) | -2.72 (-2.89, -2.55) | -2.72 (-2.78, -2.67) | -2.72 (-2.82, -2.62) |
| $\sigma^2$ (variance of spatially unstructured random effect) | 1.21 (0.82, 1.68) | - | 2.74 (0.96, 10.33) | 1.69 (1.16, 2.33) | - | 24.75 (1.80, 134.6) |
| $\sigma^2$ (variance of spatial structured random effect) | - | 0.32 (0.21, 0.45) | 36.29 (0.34, 381.1) | - | 0.50 (0.34, 0.69) | 9.54 (0.39, 16.08) |
| DIC value | 618.1 | 298.7 | 1230.0 | 673.2 | 494.9 | 495.2 |

Independent variables were standardised to have mean = 0 and standard deviation = 1; CI, Bayesian credible interval; OR, odds ratio; DIC, deviance information criterion.

fully explain the spatial clustering of TB/HIV co-infections. Maps showing the residuals of structured random effects for HIV prevalence among TB patients and for TB prevalence among people living with HIV are presented in supplementary figures (S1 Fig and S2 Fig).

## Discussion

TB and HIV co-infection is a major public health concern in Ethiopia. The present study examined the spatial distribution of TB/HIV co-infection in the whole of Ethiopia at the district level, by fitting two separate models for the prevalence of TB among people living with HIV and the prevalence of HIV among TB patients. This study found spatial clustering of HIV/TB co-infection in Ethiopia was associated with poor health care access, distance to international borders and socio-economic factors.

The prevalence of TB among people living with HIV varied strongly between districts and zones in Ethiopia, with prevalence as high as 35.9% in Etang Special zone and as low as 0.1% in Burji Special zone. The spatial heterogeneity in the prevalence of TB among people living with HIV was significantly associated with underlying variability in area-level socio-economic variables such as low wealth index, low adult literacy rate, and close distance to the international border. The observed association between the prevalence of TB and socio-economic factors has already been described in the existing literature [25–27]. A high prevalence of TB has previously been reported among people with low wealth quintile and low educational status in several countries such as South Africa [28], Kenya [29], China [30], and Brazil [31].

An interesting finding in our study was the association between the prevalence of TB among people with HIV and distance to the international border; specifically that a higher prevalence of TB was reported in the districts near to the international border as compared to districts located far from the international border. The map in our local spatial clustering analysis also showed a similar finding, in which hot-spot districts are located in Gambela, Somali, Afar, and Oromiya regions near the Ethiopia-Sudan, Ethiopia-Eritrea and Ethiopia-Kenya borders. These districts are characterised by high numbers of migrants and poor access to health care services. Our previous studies conducted in northwest Ethiopia to assess the spatial clustering of paediatric TB and MDR-TB in the general population have also detected similar hot-spot areas near the Ethiopia-Sudan and Ethiopia-Eritrea borders [32, 33]. These findings suggest that effective public health interventions are urgently needed in these remote settings to control TB transmission in the border regions.

Our study also showed that the overall prevalence of HIV infection among TB patients was 7.4%; and while this figure is considerably higher than the national HIV prevalence among the general population (0.9%) [34], it is in line with the WHO estimates of HIV prevalence among TB patients for Ethiopia (7%) [7]. A high prevalence of HIV has been consistently reported among TB patients in previous studies [35, 36]. A recent systematic review and meta-analysis conducted in sub-Saharan Africa showed a high prevalence (31.8%) of HIV among TB patients, with a gradual decline in prevalence from 33.7% in the period before 2000 to 25.7% in the period after 2010 [37].

Our study also identified wide variation in the prevalence of HIV among TB patients at the sub-national level, ranging from 0.8% in East Hararge zone to 57.1% in Liben zone. A hotspot of HIV prevalence among TB patients was observed in districts located in Amhara, Afar, and Gambela regions. This indicates that the overall national prevalence of HIV may mask the local spatial heterogeneity of the HIV epidemic and this highlights a clustered HIV transmission pattern at the district level within the country. Previous studies have also found spatial clustering of HIV prevalence among TB patients at the local level [38, 39]. Such findings suggest that the burden of HIV among TB patients should be monitored at the sub-national level

in order to effectively utilize prevention strategies and strengthen integrating services. This is particularly important in a country such as Ethiopia, where the burden of HIV and TB is high.

The spatial clustering of HIV prevalence among TB patients was significantly associated with health care access and socio-economic factors such low wealth index and low adult literacy rate. This finding is consistent with previous studies conducted in sub-Saharan Africa where a high prevalence of HIV infection was reported in areas with a low socio-economic index [40, 41]. The negative association between poor health care access and prevalence of HIV among TB patients likely relates to access to diagnostic facilities. Although HIV counselling and testing are provided in Ethiopia at all hospitals and health centres, their coverage is poor in rural and remote areas [42]. This suggests that identifying geographic areas with low wealth index and providing adequate health care services would be vital to improve local HIV identification, prevention and control programs.

The observed spatial clustering of TB and HIV co-infection in Ethiopia generally suggests that although country-level estimates are important for international comparison, they may mask important local geographic variations in the prevalence of both TB and HIV. This indicates that TB and HIV are poorly controlled in some districts. Unless effective interventions are designed and put in place, these hotspot districts could continue to be a source of TB and HIV transmission and create operational challenges for the design and implementation of collaborative TB-HIV interventions. Thus, a better understanding of the subnational variations of TB/HIV co-infection will help with the scale up of WHO-recommended TB/HIV collaborative activities to address the impact of TB/HIV co-infection and to achieve the national and global targets [31].

Our analysis had a large sample size and wide geographical coverage across Ethiopia. However, it has some limitations. First, the study was conducted based on routinely collected data that are available from the national HMIS. These data may not reflect the exact national figures as under-detection and under-reporting of cases is a potential limitation of the data. Second, since the unit of spatial analysis was districts, we were unable to estimate small scale spatial variation in HIV and TB prevalence at the community level. However, we believe that our results provide a useful operational framework for planning and setting priorities as districts in Ethiopia are the most important administrative level where local resources are allocated for each sector. Third, it would be more meaningful if we assessed the spatial distribution of TB, HIV and TB/HIV co-infection, using a multinomial model with general population as the denominator. The multinomial outcomes would be not infected, HIV mono-infection, TB mono-infection and HIV/TB co-infection. However, we were unable to do this due to the lack of population level screening data; our cohort was restricted to individuals who received secondary screening after receiving a positive diagnosis for TB or HIV. Lastly, although we used several ecological level datasets, some important covariates such as drug resistant patterns were not included in the analysis. These data were not available in the HMIS and further analysis should consider their inclusion when they become available.

## Conclusion

Our study provides evidence for geographic clustering of TB/HIV co-infection in Ethiopia and identified priority regions, zones and districts where TB/HIV collaborative activities should be strengthened to achieve greater impact with minimal additional investment.

## Supporting information

**S1 Table. Summary of independent variables, sources of data and definition of variables.** (DOCX)

**S2 Table. The prevalence of TB, HIV and TB/HIV co-infection in Ethiopia at country, region, and zone levels, between June 2016 and June 2017.**
(DOCX)

**S1 Fig. Resdiduals of structured random effects for the prevalence of HIV among tuberculosis patients in Ethiopia, June 2016 and June 2017.**
(TIF)

**S2 Fig. Resdiduals of structured random effects for the prevalence of tuberculosis among people lving with HIV (PLHIV) in Ethiopia, June 2016 and June 2017.**
(TIF)

## Author Contributions

**Conceptualization:** Kefyalew Addis Alene, Archie C. A. Clements.

**Data curation:** Kefyalew Addis Alene.

**Formal analysis:** Kefyalew Addis Alene.

**Investigation:** Kefyalew Addis Alene.

**Methodology:** Kefyalew Addis Alene, Kerri Viney, Archie C. A. Clements.

**Supervision:** Archie C. A. Clements.

**Validation:** Kefyalew Addis Alene, Kerri Viney, Archie C. A. Clements.

**Visualization:** Kefyalew Addis Alene, Kerri Viney, Archie C. A. Clements.

**Writing – original draft:** Kefyalew Addis Alene.

**Writing – review & editing:** Kefyalew Addis Alene, Kerri Viney, Hannah C. Moore, Maereg Wagaw, Archie C. A. Clements.

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
