## [Decision Letter · Decision Letter 0]

3 Nov 2019

PONE-D-19-28080

Spatial patterns of tuberculosis and HIV co-infection in Ethiopia

PLOS ONE

Dear Dr. Alene,

Thank you for submitting your manuscript to PLOS ONE. After careful consideration, we feel that it has merit but does not fully meet PLOS ONE’s publication criteria as it currently stands. Therefore, we invite you to submit a revised version of the manuscript that addresses the points raised during the review process.

We would appreciate receiving your revised manuscript by Dec 18 2019 11:59PM. To enhance the reproducibility of your results, we recommend that if applicable you deposit your laboratory protocols in protocols.io, where a protocol can be assigned its own identifier (DOI) such that it can be cited independently in the future. For instructions see: http://journals.plos.org/plosone/s/submission-guidelines#loc-laboratory-protocols

We look forward to receiving your revised manuscript.

Kind regards,

HASNAIN SEYED EHTESHAM

Academic Editor

PLOS ONE

Journal Requirements:

2. In ethics statement in the manuscript and in the online submission form, please provide additional information about the patient records used in your retrospective study. Specifically, please ensure that you have discussed whether all data were fully anonymized before you accessed them and/or whether the IRB or ethics committee waived the requirement for informed consent. If patients provided informed written consent to have data from their medical records used in research, please include this information.

3. We note you have included a table to which you do not refer in the text of your manuscript. Please ensure that you refer to Tables 3 and 4 in your text; if accepted, production will need this reference to link the reader to the Table.

Additional Editor Comments:

Minor Revision

Reviewers' comments:

Reviewer's Responses to Questions

**Comments to the Author**

1. Is the manuscript technically sound, and do the data support the conclusions?

Reviewer #1: Yes

Reviewer #2: Yes

2. Has the statistical analysis been performed appropriately and rigorously? 

Reviewer #1: Yes

Reviewer #2: Yes

3. Have the authors made all data underlying the findings in their manuscript fully available?

Reviewer #1: Yes

Reviewer #2: Yes

4. Is the manuscript presented in an intelligible fashion and written in standard English?

Reviewer #1: Yes

Reviewer #2: Yes

5. Review Comments to the Author

Reviewer #1: Comments to the authors:

Tuberculosis (TB) is the leading cause of death for individuals infected with Human immunodeficiency virus (HIV). Conversely, HIV is the most important risk factor in the progression of TB from the latent to the active status. In order to manage this double epidemic situation, an integrated approach that includes HIV management in TB patients was proposed by the World Health Organization and was implemented in many countries world-wide. To enable targeted intervention using the integrated approach, areas with high disease prevalence rates for TB and HIV need to be identified. The authors of the manuscript identified the hot and cold spot regions of HIV/TB co-infection in the districts of Ethiopia. They used a large data set to get inference on HIV/TB co-infection. In this regard, particularly in Ethiopia, the data regarding the HIV/TB co-infection are scare. This study successfully identified co-infection hot spots and associated ecological variables. The study could be used in the future to effectively manage HIV/TB co-infection in the Ethiopia. The present study is well designed, executed and the inferences drawn are well supported by the data. The manuscript is well written and understandable. Although, the concept of the study is not new, but could provide a reference for the management of TB/HIV co-infection in Ethiopia. The manuscript could be accepted for publication after incorporating the given suggestions.

Abstract:

1. The authors only mentioned hot spots for TB/HIV or HIV/TB con-infection in the districts of Ethiopia. They are advised to include cold spot regions in the abstract to distinguish between highly prevalent regions of co-infection to the least one.

Introduction:

2. The authors should also include in the introduction section, the earlier studies reporting the spatial distribution of co-infection of HIV/TB in other countries/regions. For eg. Aturinde A et al. 2019, Peres DA et al. 2019, Wei W et al. 2016.

Results:

3. The authors should include Figure S1 and Figure S2 in the main manuscript.

4. Supplementary Table S2 was incorrectly labeled as supplementary table S1. The authors should correct the labeling mistake.

Reviewer #2: The paper entitled ‘Spatial patterns of tuberculosis and HIV co-infection in Ethiopia’ by Clements et al is a comparative study of the geospatial distribution of prevalence of TB/HIV co-infection in Ethiopia. Additionally, the authors have investigated the ecological factors like health care access, distance to international borders and socioeconomic factors associated with the prevalence of HIV among people living with TB and vice versa at the district level. Furthermore, the authors found gender bias in the prevalence of TB in people living with HIV being prevalent in males than in females, whereas prevalence of HIV among TB patients was slightly lower for males than females.

The study is elaborative and very well designed with large sample size and wide geographical coverage across Ethiopia. The paper is well-written with minor syntax mistakes (which require minor corrections). Font throughout the article including the references should remain same all across. The technical procedures are correct and the results are clear-cut. Limitations of the study have been put forward by the authors very clearly. On the whole, the data is convincing and the results significantly add up to the already existing information on the particular area of research.

6. PLOS authors have the option to publish the peer review history of their article (what does this mean?). If published, this will include your full peer review and any attached files.

Reviewer #1: No

Reviewer #2: No

---

## [Author Response · Author response to Decision Letter 0]

11 Nov 2019

Manuscript Number: PONE-D-19-28080

Article Title: Spatial patterns of tuberculosis and HIV co-infection in Ethiopia

Authors: Kefyalew Addis Alene, Kerri Viney, Hannah C Moore, Maereg Wagaw, Archie CA Clements

Journal: PLOS ONE

Response to Editor’s Comments: 

1. When submitting your revision, we need you to address these additional requirements. Please ensure that your manuscript meets PLOS ONE's style requirements, including those for file naming. 

Response: We followed the PLOS ONE style templates when submitting the revised version of the manuscript. 

2. In ethics statement in the manuscript and in the online submission form, please provide additional information about the patient records used in your retrospective study. Specifically, please ensure that you have discussed whether all data were fully anonymized before you accessed them and/or whether the IRB or ethics committee waived the requirement for informed consent. If patients provided informed written consent to have data from their medical records used in research, please include this information.

Response: We have now mentioned (on page 7 line 177-178) that “This study was approved by the Australian National University Human Research Ethics Committee (protocol number 2016/218) and a letter of endorsement was obtained from the Ethiopian Ministry of Health. As this study used district-level aggregated data, informed consent was not obtained from the study participants.”

3. We note you have included a table to which you do not refer in the text of your manuscript. Please ensure that you refer to Tables 3 and 4 in your text; if accepted, production will need this reference to link the reader to the Table.

Response: Tables are now appropriately referred in the text. 

Response to Reviews’ Comments

Reviewer #1: Tuberculosis (TB) is the leading cause of death for individuals infected with Human immunodeficiency virus (HIV). Conversely, HIV is the most important risk factor in the progression of TB from the latent to the active status. In order to manage this double epidemic situation, an integrated approach that includes HIV management in TB patients was proposed by the World Health Organization and was implemented in many countries world-wide. To enable targeted intervention using the integrated approach, areas with high disease prevalence rates for TB and HIV need to be identified. The authors of the manuscript identified the hot and cold spot regions of HIV/TB co-infection in the districts of Ethiopia. They used a large data set to get inference on HIV/TB co-infection. In this regard, particularly in Ethiopia, the data regarding the HIV/TB co-infection are scare. This study successfully identified co-infection hot spots and associated ecological variables. The study could be used in the future to effectively manage HIV/TB co-infection in the Ethiopia. The present study is well designed, executed and the inferences drawn are well supported by the data. The manuscript is well written and understandable. Although, the concept of the study is not new, but could provide a reference for the management of TB/HIV co-infection in Ethiopia. The manuscript could be accepted for publication after incorporating the given suggestions. 

Abstract:

1. The authors only mentioned hot spots for TB/HIV or HIV/TB con-infection in the districts of Ethiopia. They are advised to include cold spot regions in the abstract to distinguish between highly prevalent regions of co-infection to the least one.

Response: It is now included in the revised version of the manuscript as “…hotspots were observed in districts located in Amhara, Afar, and Gambela regions, and cold spots were observed in Oromiya and Southern Nations, Nationalities, and People (SNNP) regions. …. Hotspots of TB prevalence among people living with HIV were observed in districts located in Gambela, Afar, Somali, and Oromiya regions; whereas the cold spots were observed in districts located in Amhara and Tigray regions.”

Introduction:

2. The authors should also include in the introduction section, the earlier studies reporting the spatial distribution of co-infection of HIV/TB in other countries/regions. For eg. Aturinde A et al. 2019, Peres DA et al. 2019, Wei W et al. 2016.

Response: These recent studies are now cited in the revised version of the manuscript on page 3 line 74. 

Results:

3. The authors should include Figure S1 and Figure S2 in the main manuscript.

Response: Figure S1 and Figure S2 are now included in the main manuscript on page 16 and 17.

4. Supplementary Table S2 was incorrectly labeled as supplementary table S1. The authors should correct the labeling mistake.

Response: This is now corrected.

Reviewer #2: The paper entitled ‘Spatial patterns of tuberculosis and HIV co-infection in Ethiopia’ by Clements et al is a comparative study of the geospatial distribution of prevalence of TB/HIV co-infection in Ethiopia. Additionally, the authors have investigated the ecological factors like health care access, distance to international borders and socioeconomic factors associated with the prevalence of HIV among people living with TB and vice versa at the district level. Furthermore, the authors found gender bias in the prevalence of TB in people living with HIV being prevalent in males than in females, whereas prevalence of HIV among TB patients was slightly lower for males than females. The study is elaborative and very well designed with large sample size and wide geographical coverage across Ethiopia. The paper is well-written with minor syntax mistakes (which require minor corrections). Font throughout the article including the references should remain same all across. The technical procedures are correct and the results are clear-cut. Limitations of the study have been put forward by the authors very clearly. On the whole, the data is convincing and the results significantly add up to the already existing information on the particular area of research.

Response: We greatly appreciate the reviewer’s efforts to carefully review the paper. All the errors and font styles are now corrected in the revised version of the manuscript.

---

## [Editor Report · Decision Letter 1]

21 Nov 2019

Spatial patterns of tuberculosis and HIV co-infection in Ethiopia

PONE-D-19-28080R1

Dear Dr. Alene,

We are pleased to inform you that your manuscript has been judged scientifically suitable for publication and will be formally accepted for publication once it complies with all outstanding technical requirements.

With kind regards,

HASNAIN SEYED EHTESHAM

Academic Editor

PLOS ONE

Additional Editor Comments (optional):

The Authors have revised the manuscript by addressing all the comments and questions raised by the Reviewers. New references have been added. Supplementary figures 1 and 2 have now been added to the main manuscript. Corrections to table numbering have been made.

I recommend this manuscript for publication.
---

## [Editor Report · Acceptance letter]

26 Nov 2019

PONE-D-19-28080R1 

Spatial patterns of tuberculosis and HIV co-infection in Ethiopia 

Dear Dr. Alene:

I am pleased to inform you that your manuscript has been deemed suitable for publication in PLOS ONE. Congratulations! Your manuscript is now with our production department. 

With kind regards,

on behalf of

Prof HASNAIN SEYED EHTESHAM 

Academic Editor

PLOS ONE